# Interactions between Nutrition and the “Ram Effect” in the Control of Ovarian Function in the Merino Ewe

**DOI:** 10.3390/ani12030362

**Published:** 2022-02-02

**Authors:** P. Clemens Khaiseb, Penelope A. R. Hawken, Graeme B. Martin

**Affiliations:** 1UWA School of Agriculture and Environment, The University of Western Australia, Perth 6009, Australia; Clemens.Khaiseb@mawf.gov.na (P.C.K.); penny.hawken@uwa.edu.au (P.A.R.H.); 2UWA Institute of Agriculture, The University of Western Australia, Perth 6009, Australia

**Keywords:** ovulation rate, short cycle, nutritional supplement

## Abstract

**Simple Summary:**

Clean, green, and ethical management of sheep flocks involves the use of socio-sexual stimuli (the “ram effect”) to coordinate nutritional inputs into reproductive success. However, the value of the “ram effect” is limited by three factors: (i) the proportion of the ewe flock that ovulates; (ii) ovulation rate in ewes that respond to the ram stimulus; and (iii) luteal failure after the first ram-induced ovulation, leading to short luteal phases. We tested whether these problems could be overcome by using a brief nutritional supplement (500 g lupin grain per head daily for 6 days) to stimulate ovarian activity. Lupin supplementation before ram introduction did not improve the percentage of the ewe flock that ovulates or reduce the frequency of short cycles (so will not improve the synchrony of lambing). However, lupin supplementation after ram introduction should be used to increase prolificacy.

**Abstract:**

We tested whether short-term nutritional supplementation (500 g lupin grain per head daily) would affect the response of ewes to the ram effect. Experiment 1 (end of non-breeding season): ewes were supplemented for either Days −6 to −1 relative to ram introduction (*n =* 24) or for 12 days after ram introduction (Days 11 to 22 of the ram-induced cycle; *n =* 29). Controls (*n =* 30) were not supplemented. Across all groups, 94–100% of ewes ovulated. Supplementation before ram introduction did not affect ovulation rate at the ram-induced ovulation but increased it during the ram-induced cycle (Control 1.37; supplemented 1.66; *p* < 0.05). Experiment 2 (the middle of non-breeding season): the supplement was fed for Days −5 to −1 relative to ram introduction. Again, supplementation did not increase number ovulating (Control 16/29; Supplemented 10/29) but it did increase ovulation rate at the ram-induced ovulation (Control 1.31; Supplemented 1.68; *p* < 0.05). In neither experiment did supplementation affect the frequency of short cycles. Supplementation before ram introduction did not improve the percentage of ewes ovulating or reduce the frequency of short cycles (so will not improve the synchrony of lambing). However, supplementation after ram introduction can increase prolificacy.

## 1. Introduction

The concept of clean, green, and ethical (CGE) management has become accepted as a vision for the development of animal industries because it places pressure on chemical inputs, environmental impact and animal welfare [1,2]. For the sheep industry, two primary CGE management tools are the “ram effect” for managing the timing of ovulation and therefore lambing, and “focus feeding” for the management of reproductive performance [1,3]. Focus feeding is an attractive option for extensive and subsistence production systems, where the costs of long periods of high-level feeding can be prohibitive [3].

An obvious example of “focus feeding” is the use of carefully timed, short-term supplements (e.g., with lupin grain) to enable ewes to achieve their genetic potential for fecundity by directly stimulating the metabolic processes that control ovulation (review: [4]), and by changing the balance of the negative feedback loops in the hypothalamo–pituitary–ovarian axis that control gonadotrophin secretion [5,6,7,8,9]. These processes allow FSH concentrations to be sustained during the follicular phase of the oestrous cycle, even when more follicles are growing and developing (review: [4]). The general consensus is that the critical period for nutrition to influence ovulation rate is during the 6–10-day period immediately before the next expected ovulation [7,8,9,10,11,12,13], as the last wave of follicular development progresses towards the production of ovulatory follicles (review: [4]). For such accurate timing of supplementation, the time of the next expected ovulation needs to be known and, with the ram effect, this is feasible without hormone treatment.

In the ram effect, pheromones from novel rams (or testosterone-treated castrated rams) can override the seasonal suppression of the reproductive axis and induce ovulation in anovulatory ewes (review: [14]). The ram effect induces ovulation in all of the ewes in the flock at the same time, thus offering the managerial advantage of synchronised pregnancy and lambing, and the possibility of ‘focus feeding’ to maximise reproductive outcomes [1,3]. However, the ram effect has three short-comings: (i) variation among individuals, flocks and genotypes in the percentage of ewes that ovulate [15,16]; (ii) ovulation rate at the ram-induced ovulation is highly variable, with the possibility that that ram effect itself contributes to the variation [17,18,19,20]; (iii) the corpora lutea, which follows ram-induced ovulation, often fails after about 6 days [21], so the synchrony of cycles within the flock is compromised, limiting the application of precision management, such as artificial insemination and focus feeding (reviews: [3,22,23]). All three of these problems could potentially be linked to the status of the ovarian follicles at the time of ram introduction. Considering the strong effect of acute nutritional supplements on folliculogenesis (review: [4]), it is logical that such supplements could be used to overcome the three major limitations of the ram effect:(a)The percentage of a flock that ovulates in response to the ram stimulus has long been thought to depend on the “depth of anoestrus”, a loosely defined concept that seems to involve genotype, photoperiod and time since parturition [15,16]. Follicular waves continue throughout anoestrus, although the endocrine interactions that control follicle emergence may be dampened in anoestrous ewes (review: [24]) so follicle quality and health will vary, and some follicles will not be able to respond fully to the often-rapid pre-ovulatory sequence of gonadotrophic events elicited by the ram effect (review: [15]), perhaps leading to a failure to ovulate. A role for metabolic signals in this situation is suggested by studies showing that body condition affects the number of ewes that respond to ram introduction [20,25,26], although it appears that short-term supplementation has no effect [25].(b)Long- and short-term nutrition have a major influence over ovulation rate in ewes (review: [4]) but the outcomes have been inconsistent for the ram-induced ovulation [20,25,26,27,28,29,30,31]. Some of the inconsistency is due to small numbers of ewes per treatment in some studies (a major problem with a discrete variable of very limited range), as well as the great variety in the types of nutritional treatments. For the present experiment, the most relevant report is that of Scaramuzzi et al. (2013) who used lupin grain as a supplement, as we have done in the present study, and found that it increased ovulation rate following the ram effect [25].(c)Corpora lutea that luteinizes poorly and regresses early after the ram effect [21,32,33,34] could be the outcome of ovulation of low-quality follicles, although the number and size of antral follicles do not seem to be factors [35]. The short-cycle phenomenon has also been linked to undernutrition [20,36] but we have not been able to find any reports of attempts to prevent short cycles by feeding nutritional supplements.

In the present study, we tested whether a short-term nutritional supplement with lupin grain during the preovulatory period will improve the response to the “ram effect” by increasing the percentage of ewes ovulating, increasing ovulation rate, and decreasing the frequency of short cycles. We chose lupin grain because it is highly palatable and contains little soluble starch [37] so large, acute supplements can be offered without risk of acidosis. Moreover, there is a long history of lupin treatments in the study of nutrition-reproduction interactions in sheep in many countries (e.g., [10,12,25,38,39]).

## 2. Materials and Methods

### 2.1. Ethics Statement

This experiment was carried out in accordance with the Australian Code of Practice for the Care and Use of Animals for Scientific Purposes (8th Edition, 2013) and was approved by the Animal Ethics Committee of The University of Western Australia (RA 03/100/596).

### 2.2. Area of Study and Experimental Animals

Both experiments were conducted at Allandale Farm, a field station of The University of Western Australia (UWA) that lies between 110° and 120° W and 30° and 40° S, 60 km northeast of Perth, Western Australia. The climate is Mediterranean, so the annual pasture cycle is primarily controlled by rainfall, with a growing season limited to the period from late autumn until early spring [40]. Grazing is mainly extensive with occasional hay supplements used to alleviate nutritional stress during the dry months (late October to early May). Anthelmintic treatment is routinely provided only if faecal egg counts suggest it is needed. The experimental ewes had not been treated for at least 6 months before the current studies. Both experiments were conducted under natural light, with all animals being kept outdoors. The Merino ewes were aged 3–6 years, sexually experienced, and had lambed at least once.

### 2.3. Experiment 1

The experiment was begun in mid-January, the start of the normal breeding period at the farm, just before the onset of the normal breeding season for Merinos in the southern hemisphere [41] and ended on April 30. Before the experiment commenced, all the ewes grazed dry summer pasture and were fed low-quality hay *ad libitum*.

Three specific hypotheses were tested: (i) Nutritional supplementation for six days before male introduction will increase number of ewes that ovulate in response to the ‘male effect’; (ii) Nutritional supplementation for six days before male introduction will prevent short cycles; (iii) Nutritional supplementation during the male-induced cycle will increase ovulation rate at the first oestrus (start of the second male-induced cycle).

Three days before the start of the first period of nutritional supplementation, we used transrectal ultrasonography with a real-time B-mode machine (SSD500, Aloka Co., Ltd., Tokyo, Japan) equipped with a 7.5 MHz linear array transducer, to determine the proportion of the flock that was cycling spontaneously (corpora lutea on either or both ovaries) so we could select acyclic ewes for the study (no corpora lutea).

The ewes selected for the experiment were weighed and condition-scored (BCS; scale 1 to 5, with 1 being emaciated, 2.5 medium fat and 5 very fat [42]); before random allocation among three treatments (Figure 1): Control (*n* = 32; not fed any supplement, BCS = 2.3); Lupin-supplement before male introduction (L-Pre; *n* = 23, BCS = 2.4); Lupin-supplement after male introduction (L-Post; *n* = 30, BCS = 2.4).

The L-Post group underwent the same protocol as above except that the lupin supplement was fed for 12 days, from Day 11 to Day 22 (inclusive) after male introduction. In relation to the two potential outcomes of male-induced ovulation, the timing of L-Post supplementation would have been either, (a) from the second ovulation of the male-induced ovulatory cycle for the ewes having a normal cycle only, or (b) from the third male-induced ovulatory cycle for the ewes experiencing the short cycle. The timing of this feeding regime anticipated the time the lupin supplement would affect ovulation rate [10].

The experiment was split into 2 parts:

Part 1—Prior to male introduction, the L-Pre group was run separately and fed at a rate of 500 g lupin grain (13.5 MJ/kg DM; 33% CP) per ewe per day for 6 days, while the Control and L-Post groups were run together. All groups were combined on the final day of L-Pre supplementation and testosterone-treated wethers (males castrated before puberty), fitted with harnesses and crayons, were introduced (Day 0; Figure 1). Ewes marked within the first 14 days were removed from the experiment as retrospective calculation of the timing of oestrus indicated that they were probably cyclic before the wethers were introduced. Ovarian activity was observed by laparoscopy on Day 15. Ovulation rate was determined by counting the number of corpora lutea on both ovaries [43].

Part 2—The L-Post group was separated from the Control and L-Pre groups and fed lupin grain at 500 g per ewe per day for 12 days, from Days 11 to 22 of the male-induced oestrous cycle (Figure 1). Ewes marked by the males were recorded in the morning and late afternoon every day, from Day 16 to Day 26 after initiation of the “male effect” (Day 0). The end of 14 days was also the start of the breeding programme on the farm, so the wethers were replaced with testis-intact rams fitted with different crayon colours at a 6% ratio. The colours of the crayons on the rams were changed on Days 21 and 28. Note that the rams would be perceived by the ewes as “novel” and could thus initiate a second male effect [44,45].

On Day 32 of the experiment, laparoscopy was again used to determine ovulation rate (the number of ovulations per ewe ovulating). On Day 55, trans-abdominal (3.5 MHz linear-array transducer) ultrasonography was used to determine pregnancy and to count the number of fetuses per pregnant ewe.

### 2.4. Experiment 2

The experiment was carried out during the non-breeding season, starting at the end of spring (November). We tested two hypotheses: (i) That nutritional supplementation will increase the proportion of a flock of ewes that ovulate in response to teasing in the non-breeding season; (ii) That nutritional supplementation before ram introduction will reduce the proportion of ewes experiencing the short cycle following ovulation induced by the “male effect”. We began with Merino ewes (*n* = 112) that had all lambed once before, but not in the preceding lambing season. As in Experiment 1, transrectal ultrasound was used to select acyclic ewes that were then allocated between two equal-sized groups: Control (*n =* 29) and L-Pre (*n =* 29), as shown in Figure 2.

The Control and L-Pre groups were run separately from Day −5 to Day 0 (the “male effect period”). Both groups were grazing dry pastures and fed hay ad libitum, but the L-Pre group was also fed 500 g lupin grain per ewe daily, from Day −5 to Day 0 (i.e., before male introduction). At the end of nutritional supplementation, the groups were combined, and vasectomized rams fitted with harnesses and crayons were introduced, at a ratio of 1 ram per 10 ewes.

All ewes marked within the first 14 days after ram introduction were recorded, but only data from ewes with distinct crayon marks on the rumps were retained for analysis. Crayons were replaced on Day 14 after ram introduction, and then every 7 days until Day 28. Marks were recorded every day between Days 16 and 26. Ovulation rate was determined at laparoscopy, as described above, on Day 9 for first ovulation and on Day 30 for the second ovulation in both groups. Whether an ewe experienced a normal or a short cycle was determined with countback of days from first visible display of oestrus. Throughout the experiment, ewes were weighed, and condition scored every 7 days, as described for Experiment 1. All groups (Control; L-Pre; L-Post) had a mean BCS of 2.4.

### 2.5. Statistical Analyses

We have based the analysis of treatment effects only on data collected from ewes that were deemed to have responded to the “male effect”, as evidenced by the occurrence of oestrus between Days 16 and 26 after male introduction. Chi-squared tests (χ^2^) in 2 × 2 contingency tables were used to compare the number of corpora lutea per ewe ovulating (ovulation rate) and the numbers of ewes defined as ovulating in response to the ram effect.

## 3. Results

### 3.1. Experiment 1

Live weight (Table 1) did not differ among the three groups at the beginning or at the end of the experimental period (*p* > 0.05); similarly, changes in weight were not affected by treatment (*p* > 0.05).

The data for oestrus, ovulation and ovulation rate, and the proportions of ewes with short and normal cycles after the ram-induced ovulation, are also shown in Table 1. In all three groups, almost all ewes ovulated and there was no effect of treatment on the proportion of ewes responding to the male effect. The percentage of ewes ovulating closely followed the percentage of ewes showing oestrus, and more ewes ovulated than displayed oestrus in all three groups.

The ovulation rate tended (*p* > 0.05) to be higher in the L-Pre group than in the Control at the first ovulation but was significantly higher in the L-Post group than in the Control (*p* < 0.05) on Day 32, at the second ovulation for ewes with a normal cycle and at third ovulation for ewes with short and normal cycle. Following the male-induced ovulation, the pattern of oestrus was similar for L-Pre and Control (*p* > 0.05). In all groups, more ewes were marked between Days 19 and 20 than between Days 21 and 29, indicating a majority of normal cycles. The proportion of ewes with a normal cycle was greater than the number of ewes with the short cycles in the L-Post group (*p <* 0.05). This difference was not seen in the Control and L-Pre groups (*p* > 0.05). Consequently, 67% of the ewes experienced a normal cycle in the L-Post group, exceeding the percentages in the C and L-Pre groups (*p* < 0.05). Pregnancy rates (Table 1) did not differ among the groups. There were significantly more twin foetuses (and thus less singles) in L-Post than in the other two groups.

### 3.2. Experiment 2

Live weights were similar for the two groups throughout the experimental period (*p > 0.05*; data not shown). In the Control group, the same proportion 16/29 (55%) of ewes exhibited oestrus as ovulated. In contrast, only 10/29 (34%) ewes ovulated in the L-Pre group, largely due to 21% of ewes failing to show oestrus; these ewes were subsequently excluded from the analysis.

The data for ovulation rate and oestrus on Days 10 and 30 after the “male effect” are also shown in Table 2. Ovulation rate was significantly greater in L-Pre ewes than in Controls on Day 10 (*p* < 0.05) but declined by the second ovulation on Day 30 to be similar to the Control value. As shown in Table 2, less than 50% of the ewes experienced a normal cycle in both groups, but more were observed in the Control than in the L-Pre group (*p* < 0.05). The ram-induced ovulation resulted in similar patterns of oestrus for L-Pre and Control (*p* > 0.05). In both groups, more ewes were marked between Days 16 and 18 (a normal cycle length following the ram effect) than between Days 23 and 24 (ewes experiencing a short cycle after the ram effect followed by a normal cycle). More ewes (16/29; 55%) showed oestrus in the Control group than in the L-Pre group (15/29; 51%), but the difference was not significant (*p* > 0.05).

## 4. Discussion

### 4.1. Proportion of the Flock Induced to Ovulate by the Ram Effect

The hypothesis that a nutritional supplement would increase the proportion of the ewe flock ovulating was effectively only tested in Experiment 2. It was clear that the lupin supplement provided no benefit, confirming other studies with short-term supplementation [25,26,27,28,29,30,31], although some of those same studies [25,26,27] suggest that good long-term nutrition, leading to better body condition, could improve the outcome and thus offer a valuable management option. In Experiment 1, the hypothesis could not be tested because ovulation was observed in all ewes in the L-Pre and L-Post groups and in 94% of Control ewes. It appears that almost all ewes responded to the ram effect, perhaps along with a small contribution from spontaneous ovulations (as evidenced by 10% of the flock cycling when the experimental animals were selected just before the experiment). The time of the season probably played a major role—Experiment 1 was begun around the expected onset of the normal, photoperiod-induced breeding season, when we would expect an increase in the number of spontaneously ovulating ewes.

It might be worthwhile testing the hypothesis again in breeds that are more seasonal than Merinos, where the response to the ram effect tends to be less robust [15,16]. Nevertheless, the most logical interpretation is that the outcome of the ram effect, in terms of whether the ewe ovulates or not, is not determined at the level of the ovary but at the level of the brain-pituitary systems that control gonadotrophin secretion [46,47,48,49].

### 4.2. Ovulation Rate in Ewes following the Ram Effect

On the other hand, ovarian follicular activity was stimulated by the nutritional supplement, as evidenced by the increase in ovulation rate in both experiments. The most interesting observations were those from Experiment 2 where the supplement increased ovulation rate at the first, ram-induced ovulation, but ovulation rate subsequently returned to control values after the supplement was withdrawn. The observations from both experiments confirm the importance of nutritional supplementation during the immediate preovulatory period, when it lengthens the lifespan of the last non-ovulatory follicle and seems to delay the initiation of the next follicular wave, thus allowing more follicles of larger size to develop [4,5,6,7,8,9,11]. It is notable that Scaramuzzi et al. [25] did not detect an effect of lupin supplementation on ovulation rate after the ram effect. This disagreement can be explained by ewe genotype (Merinos being less seasonal than Ile-de-France) and by our use of larger treatment groups. Small groups are compatible with intensive endocrine measurements, a key component of the protocol used by Scaramuzzi et al. but make it difficult to detect effects on discrete variables such as ovulation rate and short cycles.

The outcome in Experiment 1 was unexpected and, in hindsight, could be attributed to the limited group size or the time of year. The effect of supplements on ovulation rate is considered to be more robust in ewes that had previously been grazing in a poor quality pasture [50], a situation relevant to Experiment 1, as the ewes had been run on a dry summer pasture and, although they were given hay *ad libitum*, it was of poor quality. Thus, as grazing conditions deteriorated with the progress of autumn, the second period of supplementation was more likely to produce a significant response to the lupin treatment. In any case, the first, ram-induced ovulation is not accompanied by behavioural oestrus unless the ewes are primed with progestagen [15,32], so it is probably best that producers continue nutritional supplementation until after the second and third ovulations as a way to improve reproductive efficiency.

### 4.3. Frequency of Short Cycles following the Ram Effect

Despite follicle development being stimulated by the lupin supplement, as evidenced by the increase in ovulation rate, the supplement did not reduce the proportion of ewes experiencing short cycles. Clearly, luteal failure is not caused by metabolic limitations to follicle development in the pre-ovulatory period, a conclusion that is consistent with the evidence that an inadequate follicle pool is not responsible for the failure of the ram-induced corpus luteum [35]. However, the possibility remains that poorly developed follicles are ovulated, especially when there is only brief ram-induced gonadotrophic stimulus before the preovulatory surge intervenes [32,45,48,49]. Indeed, there is good evidence of poor luteinization after the “ram effect” [51,52]. Equally, there is good evidence that the uterine luteolytic signals play a role in short cycles, as evidenced by studies based on hysterectomy [34,53,54,55,56]. We seem to be gaining an understanding of the cause of short cycles following ram-induced ovulation but need much more progress if we are to provide a solution relevant to producers, a critical problem is that it prevents maximization of the value of ram-induced ovulation in flock management [22].

### 4.4. Reproductive Performance following the Ram Effect

As shown in Experiment 1, nutritional supplementation for 12 days during the oestrous cycle improved reproductive performance, as evidenced by the increased percentage of twin foetuses observed at scanning, supporting the earlier observation of increased ovulation rate. Again, it is clear that, in addition to feeding for 6 days before the “ram effect”, supplementation should continue until after subsequent ovulations. In other words, rather than aiming for precise timing, it would be better for sheep producers to ensure that the duration of the feeding is extended so that it also covers the 6-day period leading to the second ovulation (or the third, in the case of ewes experiencing a short cycle). Producers in Mediterranean regions can be confident of this approach because the effect of single, short-term nutritional supplements on ovulation rate diminishes from spring to summer, then recovers in early autumn [57].

## 5. Conclusions

A short-term nutritional supplement before ram introduction has no benefit with respect to the proportion of the ewe flock that ovulates, or the probability of a short cycle (thus offering no improvement in the synchrony of lambing). A nutritional supplement before ram introduction can improve ovulation rate at the first ram-induced ovulation, but this ovulation rarely leads to pregnancy as there is no oestrus behaviour. In contrast, prolificacy can be improved by nutritional supplementation over a period of 14–20 days after the initial ram effect, as this increases ovulation rate at the second ovulation, when conception is likely.

## Figures and Tables

**Figure 1 animals-12-00362-f001:**
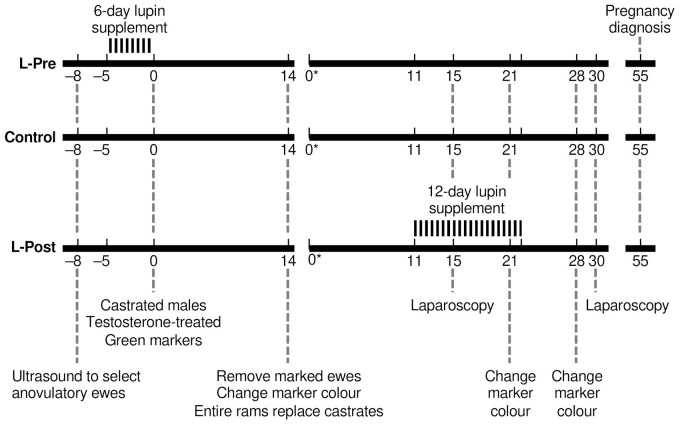
The protocol for Experiment 1. The Control and L-Post groups were run together, and separated from the L-Pre group, until Day 0 (the final day of supplementation for L-Post, and the day males were introduced). Day 0* is the supposed time of ovulation for the second male-induced cycle.

**Figure 2 animals-12-00362-f002:**
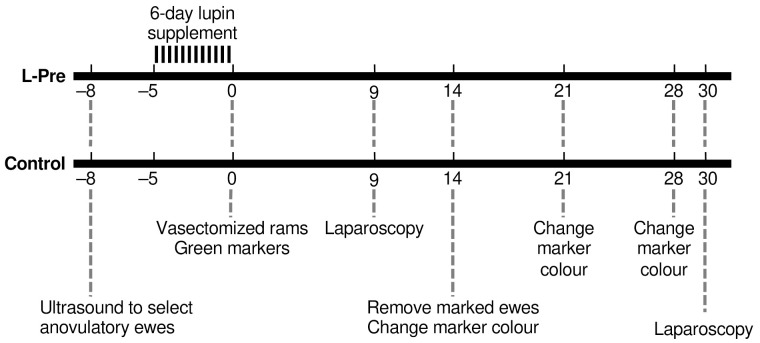
The protocol for Experiment 2. The Control and L-Pre groups were separated until Day 0 (the day males were introduced).

**Table 1 animals-12-00362-t001:** Data for Merino ewes induced to ovulate using the “male effect” at the onset of the breeding season. Data for ovarian activity were collected at laparoscopy on Day 15 for L-Pre (lupin supplement fed before the introduction of males) and Day 30 for L-Post (lupin supplement fed after the first male-induced ovulation). Data for pregnancy diagnosis were collected using ultrasound on Day 55 after male-induced ovulation. Different superscripts indicate significant differences (*p* < 0.05).

	Control (*n* = 32)	L-Pre (*n* = 23)	L-Post (*n* = 30)
Live weight Day 1 (kg)	47.1 ± 0.85	45.9 ± 0.91	47.7 ± 0.96
Live weight Day 32 (kg)	48.0 ± 0.82	46.1 ± 0.90	48.0 ± 0.85
Ewes ovulating (%)	94 (30/32)	100 (23/23)	100 (30/30)
Ovulation rate	1.37 ^a^	1.48 ^a^	1.66 ^b^
Ewes showing a short cycle (%)	34 (10/30) ^a^	39 (9/23) ^a^	23 (7/30) ^b^
Ewes showing a normal cycle (%)	41 (12/30) ^a^	48 (11/23) ^a^	67 (20/30) ^b^
Ewes showing oestrus (%)	75 (22/30)	87 (20/23)	90 (27/30)
Ewes pregnant at scanning (%)	72 (22/30)	83 (19/23)	73 (22/30)
Ewes with a single foetus (%)	74 (16/22)	63 (12/19)	45 (10/22)
Ewes with twin foetuses (%)	26 (6/22) ^a^	37 (7/19) ^a^	55 (12/22) ^b^

**Table 2 animals-12-00362-t002:** Reproductive variables in Merino ewes induced to ovulate using the “male effect” during the anoestrus season. Ovulation data were collected at laparoscopy on Day 10 and Day 30. Different superscripts indicate significant differences (*p* < 0.05).

	Control (*n* = 29)	L-Pre (*n* = 29)
Ewes ovulating	16 (55%)	10 (34%)
Ovulation rate (Day 10)	1.31 ^a^	1.68 ^b^
Ewes showing oestrus	16 (55%)	15 (51%)
Ewes showing a short cycle	3 (10%) ^a^	5 (17%) ^a^
Ewes showing a normal cycle	13 (45%) ^a^	10 (34%) ^b^
Ovulation rate (Day 30)	1.40 ^a^	1.35 ^a^

## Data Availability

None of the data were deposited in an official repository, but information can be available upon request and the MSc thesis of P.C.K. is available online.

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
