# Peer review of "Interactions between Nutrition and the “Ram Effect” in the Control of Ovarian Function in the Merino Ewe"

_animals, 2022, doi:10.3390/ani12030362_

Round 1

Reviewer 1 Report

I know very well the studies of Prof. Martin and his team about the concept of clean, green and ethical management. My opinion is that their  studies are original and valuable as an alternative of pharmaceutical control of estrus of sheep. The present study continues  the aim of team works and it is original and add new knowledge about flushing and ram effect and their combined effect. My opinion is that the present study could be accepted in present form.  But if the authors agree with my notes they can include in their work.

Questions and notes:

  1. Did you do dehelmintization and test of parasite infestation of experimental ewes? If yes, you could note in article.
  2. Why the animals in Experiment 1 were not equal numbers?
  3. Did you do analysis of contents of grass, hay and lupin grain? It is interesting to know the crude protein and energy values? If yes, you could note in articles.
  4. It would be good, in conclusions to add recommendations to practice, which protocol for flushing is better to use in the beginning of  breeding season - flushing before ram introduction or after. From the results of the study, my opinion is, that flushing after ram introduction is better.  

Author Response

1. Did you do dehelmintization and test of parasite infestation of experimental ewes? If yes, you could note in article.

Response: This information has now been added to Section 2.2.

Why the animals in Experiment 1 were not equal numbers?

Response: ewes were eliminated on the basis of retrospective evidence of spontaneous cyclicity, as stated on:

Page 4: “Ewes marked within the first 14 days were removed from the experiment as retrospective calculation of the timing of oestrus indicated that they were probably cyclic before the wethers were introduced.”

Page 8: “We have based the analysis of treatment effects only on data collected from ewes that were deemed to have responded to the ‘male effect’.”

The outcome of the elimination was not evenly distributed among the groups. No change to the manuscript.

Did you do analysis of contents of grass, hay and lupin grain? It is interesting to know the crude protein and energy values? If yes, you could note in articles.

Response: no analysis was done. However, it is an important consideration. This nutritional environment and the lupin supplement have a long history of scientific study. We have added information for lupin grain (13.5 MJ/kg1 DM; 33% CP) as well as a reference [37] to the end of the Introduction and to Sections 2.2 and 2.3.

It would be good, in conclusions to add recommendations to practice, which protocol for flushing is better to use in the beginning of  breeding season - flushing before ram introduction or after. From the results of the study, my opinion is, that flushing after ram introduction is better.  

Response: we agree. We have added this point to the Simple Summary, the Abstract, and Section 5 (Conclusion).

Reviewer 2 Report

The manuscript brings new information concerning nutritional supplement before and during breeding season and during non-breeding season. The study design is rationale, the methods were properly chosen, the results are clearly presented and the conclusions are justified by them. In my opinion the results are very interesting and valuable for the sheep breeders interested in improvement of reproductive performance.

The only minor remark is the mistake at the 8th page (second sentence) – should be unless

Taking into consideration all the above in my opinion the manuscript is suitable to be published in Animals.

The main question addressed is if ovarian activity can be influenced by special (additional) nutrition together with the presence of the ram during and out of the breeding season?

In my opinion the study is interesting because of importance for the sheep breeders, nutritionists and vets. The study design, methodology are not very sofisticated but due to the lack of (small number) of such practical trials it is worth publishing. The authors presented data of the ovarian activity and reproductive performance of sheep included. The description of the feeding regime, its duration, time of the entrance of a ram into the flock can be used by the other researchers and owners as it was described or can be modified for another studies.

Because the number of the lambs and their vitality are very important for the sheep breeders the study aimed to improve such parameters are important from pracatical point of view.

I would not suggest any improvements because the study design is clear and the used methods were sufficient.

I have no more comments concerning tables and figures. 

Author Response

Response: this mistake was actually on Page 11. Now corrected.

Reviewer 3 Report

The manuscript with id animals-1566044 evaluates the CGE management for controlling the reproductive processes in Merino ewes. The study is interesting and the manuscript is well written and comprehensive. However, the reader may wonder what is the novelty of the present study, the breed, the season or what? Please clarify the novel objectives and results, if there are, of the study, as lupin supplement has been evaluated many times during the last 5 decades. Which is the difference with previous studies (e.g with reference no 25)?

Why did you choose lupin for focus feeding, please add some information about this. 

In Table 1 please add for each percentage the exact number of ewes for each case (e.g. ??/32 for control group) and write ovulation rate as mean+SE. Ovulation rate between the three groups should have been analysed by means of ANOVA.  Also, there are presented results that have not been written in Statistical Analysis (See Statistical analysis and results presented in both Tables)

In Conclusion add the sentence referring to the evaluation of the present results as it was discussed in the previous paragraph (i.e. ‘Again, it is clear that, in addition to feeding for 6 days before the ‘ram effect’, supplementation should continue until after the third ovulation).

Author Response

1) The reader may wonder what is the novelty of the present study, the breed, the season or what? Please clarify the novel objectives and results, if there are, of the study, as lupin supplement has been evaluated many times during the last 5 decades. Which is the difference with previous studies (e.g with reference no 25)?

Response: Agreed. We have added a commentary in Section 4.2, where it is most relevant.

2) Why did you choose lupin for focus feeding, please add some information about this.

Response: this question has now been addressed at the end of the Introduction. Extra references [37–39] were needed.

3) In Table 1 please add for each percentage the exact number of ewes for each case (e.g. ??/32 for control group) and write ovulation rate as mean+SE. Ovulation rate between the three groups should have been analysed by means of ANOVA.

Response: We agree that this Table was inadequate. We have added the ratios, as requested, and slightly restructured the table so it flows more logically.

4) Write ovulation rate as mean+SE. Ovulation rate between the three groups should have been analysed by means of ANOVA.

Response: As we stated in Section 2.5: “The number of corpora lutea per ewe ovulating (ovulation rate) was analysed using Chi-squared tests (χ2) in a 2 x 2 contingency table.” This is the correct analysis for a discrete variable with very narrow range. Ovulation rate data are never normally distributed so an SE is never appropriate. No change to the manuscript.

5) There are presented results that have not been written in Statistical Analysis (See Statistical analysis and results presented in both Tables)

Response: agreed. We have modified Section 2.5, including deletion of the text “Oestrus distributions for short versus normal cycles were determined using histograms with fits and groups using Minitab 15”.

5) In Conclusion add the sentence referring to the evaluation of the present results as it was discussed in the previous paragraph (i.e. ‘Again, it is clear that, in addition to feeding for 6 days before the ‘ram effect’, supplementation should continue until after the third ovulation).

Response: Agreed (also requested by Reviewer 1). We have clarified this point in the Simple Summary, the Abstract, and Section 5 (Conclusion).

Reviewer 4 Report

The manuscript reports on a study on the Interactions Between Nutrition and the 'Ram Effect' in The Control of Ovarian Function in the Merino Ewe.
The study is interesting because the subject is treated and it is always up to date due to the importance it has for sheep breeding. Furthermore, saving both energy resources and animal treatments are essential factors in the ecological transition of farms.
The introduction is too long and some of the bibliographies cited do not exactly agree with what is written.
The aim must be more centered on what you want to do, it must not generally leave the hypothesis that you want to solve much bigger things.The materials and methods are clear and the tables on the experimental groups also help the reader to understand the study.
The results are consistent with the study. The tables are understandable but the data shown must be expressed in the same way in the two tables.
The discussion is well organized but many times it does not explain why food supplementation did not bring about the expected response.
The conclusions must be improved and must be consistent with the purpose.
The manuscript after the suggested modifications can be published in the journal.

Author Response

1) The introduction is too long.

Response: We are generally strongly in favour of short introductions but, in this case, we need to justify three major hypotheses. They are all superficially related, but they are independent in terms of science background. As a consequence, we need to present logical arguments for each one. That said, we have looked at the Introduction and edited many areas to reduce the number of words. Unfortunately, Reviewer 3 wanted us to justify the use of lupin grain, so extra words were added.

2) Some of the bibliographies cited do not exactly agree with what is written.

Response: This is always a frustrating danger when numbered references are used! Specific examples would have been useful. Nevertheless, we have carefully gone through the manuscript and searched for such problems. The only major issue we found was an error when we quoted the outcome for ovulation rate in Reference 25. We are very pleased to have fixed it.

3) The aim must be more centered on what you want to do, it must not generally leave the hypothesis that you want to solve much bigger things.

Response: We agree. At the end of the Introduction, we have deleted the words “in the pursuit of a practical resolution of these problems for Merino ewes”, thus focussing n the hypothesis.

4) The tables are understandable but the data shown must be expressed in the same way in the two tables.

Response: Excellent request. Reviewer 3 also wanted more detail. Table 1 has now been clarified and it matches Table 2.

5) The discussion is well organized but many times it does not explain why food supplementation did not bring about the expected response.

Response: we have made some modifications to the Discussion, particularly around the issue of ovulation rate (Section 4.2). We appreciate the desire for explanations but, in the absence of explanatory measurements, there is a risk of excessive speculation.

6) The conclusions must be improved and must be consistent with the purpose.

Response: Agreed. A similar request was also made by Reviewers 1 and 3. We have clarified this point in the Simple Summary, the Abstract, and Section 5 (Conclusion).